# Effect of Proofing on the Rheology and Moisture Distribution of Corn Starch-Hydroxypropylmethylcellulose Gluten-Free Dough

**DOI:** 10.3390/foods12040695

**Published:** 2023-02-06

**Authors:** Duqin Zhang

**Affiliations:** Institute of Cereal & Oil Science and Technology, Academy of National Food and Strategic Reserves Administration, Beijing 100037, China; zdq@ags.ac.cn

**Keywords:** proofing, gluten-free dough, rheology, moisture distribution, soluble carbohydrates

## Abstract

Dough rheology, mainly enabled by gluten in the traditional dough, determines the end-products’ quality, particularly by affecting gas production and retention capacities during proofing. Gluten-free dough has quite different rheological performance compared with gluten-containing dough. To deepen the understanding of gluten-free dough, variations of rheology and moisture distribution of corn starch-hydroxypropylmethylcellulose (CS–HPMC) gluten-free dough in the process of proofing were studied. Significant differences were found in terms of soluble carbohydrate composition, moisture distribution, and rheology. Arabinose, glucose, fructose, and mannose were the main composition of soluble carbohydrates in CS–HPMC dough, out of which glucose was preferentially utilized during proofing. Non-freezable water content and third relaxation time decreased from 44.24% and 2171.12 ms to 41.39% and 766.4 ms, respectively, whereas the amplitudes of T_23_ increased from 0.03% to 0.19%, indicating reduced bounded water proportion and improved water mobility with proofing time. Frequency dependence and the maximum creep compliance increased, whereas zero shear viscosity reduced, suggesting decreased molecular interactions and flowability, but improved dough rigidity. In conclusion, the reduced soluble carbohydrates and improved water mobility decreased molecular entanglements and hydrogen bonding. Furthermore, yeast growth restricted a large amount of water, resulting in declined flowability and increased rigidity.

## 1. Introduction

Three major stages, including mixing, proofing (i.e., fermentation), and thermal setting (i.e., baking/steaming), were identified in the process of bread making [1]. Among the three stages, proofing is the process of biochemical modification of the dough matrix by microorganisms (especially yeast and lactic bacteria) and their metabolites [2]. In this stage, a large amount of complex microbial activity was not limited to gas (CO_2_) production; it was the major reason for rheology variation and end-products quality (e.g., the final loaf volume, crumb structure, texture, and flavor of the bread) [1,3,4].

As is known to all, gluten plays a crucial role in imparting viscoelasticity properties to a traditional dough matrix [5]. Rheological variation of gluten-containing dough in the proofing stage was widely studied [6]. For instance, the expansion of gluten-containing dough resulting from *Saccharomyces cerevisiae* or baker’s yeast action was responsible for the increase of porosity and the change of alveolar structure stability [3]. The mixing step made the flour constituents homogenized with air bubbles and created intra- and intermolecular associations between flour constituents and moisture; while during proofing, nutrients, air, and moisture were utilized by microbes to produce metabolites, such as ethanol, organic acids, exopolysaccharides, etc., exerting great influence on gluten networks and eventually affecting dough rheology [7,8]. However, gluten has been identified as the cause of celiac disease, leading to the damage of intestinal mucosa and malabsorption of several important nutrients [9]. In gluten-free diets, gluten was replaced by hydrocolloids, which were able to bind water and create non-gluten networks stabilized by inter- and intra-hydrogen bonds, increasing the viscoelasticity of the dough matrix [10]. Apart from hydrocolloids, starch, ranging from 60% to 90%, was the most important component and played a crucial role in the gluten-free diet as well [11].

In the initial proofing stage, the readily fermentable sugar concentration in dough is probably limiting; thus, starvation conditions were suffered for yeast [12]. The imbalance situation between yeast consumption and starch hydrolysis contributed to a rapid depletion of soluble carbohydrates [13]. Subsequently, yeast cells adjust their growth rate to nutrient availability and use a diverse array of compounds as carbon sources (e.g., aerobic utilization of ethanol) and are capable of expressing catabolic enzymes (including α-amylase, β-amylase, protease, etc.) of many different pathways [14]. According to Shehzad et al. (2010) [13], the action of yeast on low molecular weight carbohydrates results in the production of CO_2_, which increases dough volume and contributes to overall shape and crumb texture development. Furthermore, the partial hydrolysis of starch and protein by yeast during proofing softened the gluten-containing dough (especially the gluten network) and decreased the dough stability [15]. However, different from gluten-containing dough, the study of the rheological variation and the moisture distribution (which could reflect yeast proliferation and metabolism) of the gluten-free dough in the absence of gluten during proofing were insufficient. Therefore, to deepen our understanding of the rheology changes of the gluten-free dough in the process of proofing, this work investigated the soluble carbohydrate composition, water morphology and mobility of the corn starch-Hydroxypropylmethylcellulose (HPMC) gluten-free dough on a microscopic scale; then, we studied the rheological changes of the CS–HPMC gluten-free dough on the macroscopic scale during proofing through dynamic rheological characterization and creep and recovery measurements. This work could contribute to the understanding of the rheological changes that take place during the proofing process and facilitate the development of breads with improved quality and consistency.

## 2. Materials and methods

### 2.1. Materials

The corn starch (CS) was purchased from Yuanye Biotechnology Co., Ltd. (Shanghai, China). HPMC was obtained from Sigma-Aldrich LLC (Merck KGaA, Darmstadt, Germany). The number average molecular weight of HPMC is 120,000 with 21% methoxyl and 5% hydroxypropyl substitution degrees, respectively. All the other chemical reagents used in this study were of analytical grade and purchased from Sinopharm Chemical Reagent Beijing Co., Ltd. (Beijing, China).

### 2.2. Dough Preparation

To prepare CS–HPMC gluten-free dough, 2.0% (*w*/*w*) HPMC, 2.0% (*w*/*w*) high active dry yeast (Angel Yeast Co., Ltd., Yichang, China), and 71.70% (*w*/*v*) water were added to CS [16]. The HPMC and high active dry yeast were dissolved in the water at 30 °C first, and then added to CS and mixed together at low speed (80 rpm) for 15 min in a Hobart mixer A-120 (The Hobart Manufacturing Company, Tory, OH, USA). The prepared CS–HPMC doughs were immediately placed in a temperature and humidity chamber (BSC-150, Boxun medical biological Instrument Co., Ltd., Shanghai, China) and fermented under the controlled conditions of 35 °C and 80% relative humidity for 30, 60, and 90 min, and, respectively, marked as D-30, D-60, and D-90. The CS–HPMC dough fermented for 0 min was used as control and marked as D-0.

Approximate 10 g of the CS–HPMC dough was divided, allowed to rest at 25 ± 0.1 °C for 3 min, and used for dynamic rheological characterization, and creep and recovery measurement. A total of 50 g of the CS–HPMC dough was accurately divided and applied for water mobility analysis.

### 2.3. Proofing Curves Determined by Rheofermentometer

To investigate changes of dough volume and its gas production capacity in the process of proofing, a Rheofermentometer (Rheo F4, Chopin Technologies, Paris, France) was applied following the procedure described in reference [17]. In brief, 315 g CS–HPMC dough was added to the fermentation basket of the gas meter with a 500 g cylindrical weight, followed by covering the vat immediately, which was fitted with an optical sensor. The proofing chamber was hermetically closed and the measuring series was started at 35 °C for 180 min. The proofing curves including dough height and gas production curves were record.

### 2.4. Soluble Carbohydrate Composition

Variation of the soluble carbohydrate composition in the CS–HPMC gluten-free dough before and after proofing were determined following the procedure described in reference [18]. 3 g of CS–HPMC dough was dispersed in 30 mL of ethanol and boiled for 20 min. In the cooling stage, the mixture was centrifuged at 4000× *g* for 15 min. The supernatant was filtered using Waterman No. 42 filter paper and concentrated by vacuum evaporator at 45 °C. The concentrated solutions were adjusted to 2 mL constant volume by adding ethanol. Monosaccharide contents were quantified by high-performance anion-exchange chromatography with pulsed amperometric detection (HPAEC-PAD) following acid hydrolysis with 4 M trifluoroacetic acid at 121 °C for 2 h. Chromatography of the samples was carried out in a Diones ICS-3000 Bio-LC system, using a CarboPac PA 20 column (250 mm × 4 mm) in combination with a CarboPac guard column (Diones Corp. Sunnyvale, CA, USA). After being filtered through a 0.22 um filter membrane, 20 μL of the abovementioned solution was injected. All analyses were carried out at 30 °C and a flow rate of 1.0 mL/min with 75% (*v*/*v*) acetonitrile as eluent solution.

Standard solutions of monosaccharide (0.2–20 μg/mL) were prepared to confirm the linearity of the detector and determine the relative response factor. Retention time: arabinose 4.8 min, glucose 6.3 min, fructose 6.9 min, and mannose 7.6 min (as shown in Appendix A).

### 2.5. Water Morphology Determined by DSC

Water morphology (freezable and non-freezable water) of the CS–HPMC gluten-free dough was determined by DSC apparatus (Q200 TA Instruments, New Castle, DE, USA) according to the procedure in reference [19] with slight modifications. Briefly, a total of 20 mg of CS–HPMC dough was placed into the DSC special-use aluminum pans. The pans were hermetically sealed and cooled from 25 °C to −40 °C at the rate of 5 °C/min using liquid nitrogen, and then heated to 40 °C at the rate of 5 °C/min. A hermetically sealed empty pan was used as a reference. The enthalpy (ΔH) of the melting peak was determined with Universal Analysis software (TA Instruments). The freezable water content (C_F_, %) was calculated by the formula as follows:(1)$$\mathrm{CF}\left(\%\right)=\frac{\Delta H}{{\Delta H}_{0}\;}\times 100\%$$
where ΔH was the enthalpy of the melting peak of the endothermic curve, J/g; ΔH_0_ was the enthalpy of the melting peak of pure water, which was 334 J/g in this study. Each measurement was performed in triplicate.

### 2.6. Water Mobility Analysis

Water mobility analysis of CS–HPMC gluten-free dough was carried out using low-resolution ^1^H nuclear magnetic resonance (NMR; Low-resolution MesoMR Spectrometer, Niumag, Shanghai, China) operating at a ^1^H resonance frequency of 23 MHz. In brief, a total of 50.00 g dough was placed at the center of a permanent magnetic field in the radio-frequency coil position immediately after kneading. The spin–spin relaxation time, T_2_, was measured and recorded by applying the Carr–Purcell–Meiboom–Gill pulse sequence. The typical pulse parameters were: 17 μs dwell time; 400 μs echo time; 1500 ms recycle time; 5000 echo counts; and four scan repetitions [20]. All measurements were performed in triplicate.

### 2.7. Dynamic Rheological Characterization

A controlled stress rheometer (Physica MCR301; Anton Paar, Graz, Austria) was used to measure the dynamic rheological characteristics of the CS–HPMC dough [16]. The experiment was performed in a geometry parallel-plate with a 25 mm diameter and a gap of 1 mm. Each CS–HPMC dough sample was placed between the plates after being mixed and resting for 25 min. Silicone oil was used to cover the rim of the dough in order to prevent water evaporation during the test.

#### 2.7.1. Dynamic Strain Sweep

The linear viscoelasticity region (LVR) of the CS–HPMC gluten-free dough was determined by dynamic strain sweep, which was performed over a strain range of 0.01–10% at an angular frequency of 10 s^−1^ and 25 °C.

#### 2.7.2. Dynamic Frequency Sweep

Frequency sweep experiments were performed on the CS–HPMC gluten-free dough from 0.1 to 100 s^−1^ at 0.1% strain and 25 °C. The functional relationships between G′ (storage modulus) and angular frequency (ω) were recorded. By fitting the frequency sweep data into the power law model (as shown in Equation (2)), values of z′ (reflects type of molecular interactions) and K′ (reflects strength of molecular interactions) were calculated.
G′ = K′(ω)^z′^
(2)

#### 2.7.3. Temperature Sweep

The temperature sweep was operated after dynamic frequency sweep. During sweep, changes of G′ and G″ of the dough samples were recorded as the temperature increased from 25 to 90 °C at a heating rate of 5 °C/min under the condition of 1 Hz constant frequency and 0.05% strain. The complex shear modulus (|G^*^|) and the loss factor (tan δ) were recorded.

### 2.8. Creep and Recovery Behavior

A controlled stress rheometer (Physica MCR301, Anton Paar, Austria) equipped with a Peltier temperature control system at 25 ± 0.1 °C was used to measure the creep and recovery behavior of the CS–HPMC gluten-free dough, which was analyzed using a parallel-plate geometry with a 25 mm diameter and a gap of 2 mm [20]. The prepared dough was placed on the lower parallel plate. After lowering the upper parallel plate, the excess dough sample was trimmed and sealed with a silicone oil to prevent the dough from drying. The creep phase was recorded at a shear stress of 250 Pa for 300 s, and then the recovery phase was recorded for 300 s at a stress of 0 Pa. The deformation of the CS–HPMC gluten-free dough during the creep and recovery phase can be quantified by fitting the experimental data into the Burgers model [21].

For the creep phase, the Burgers models are:J_c_(t) = J_0_ + J_m_(1 − exp(−t/λ)) + t/η_0_(3)

In Equation (3), J_c_ (Pa^−1^) corresponds to creep compliance, J_0_ (Pa^−1^) corresponds to instantaneous compliance, J_m_ (Pa^−1^) corresponds to viscoelastic compliance, λ (s) corresponds to mean retardation time, t (s) corresponds to test time (t = 0), and η_0_ corresponds to shear viscosity.

For the recovery phase, the Burgers models are: J_r_(t) = J_max_ − J_0_ − J_m_(1 − exp(−t/λ))(4)

In Equation (4), J_r_ (Pa^−1^) is the creep compliance during recovery phase, t(s) corresponds to the test time (t = 300 s).

For steady state, the dough is in the equilibrium state after the recovery phase. The Burgers models are:J_v_/J_max_ = J_m_/J_max_(5)
J_e_/J_max_ = (J_max_ − J_m_)/J_max_(6)

In Equations (5) and (6), J_max_ corresponds to the maximum creep compliance, J_v_/J_max_ corresponds to the relative viscous part of the maximum creep compliance, and J_e_/J_max_ corresponds to the relative elastic part of the maximum creep compliance.

### 2.9. Statistical Analysis

One-way analysis of variance followed by Duncan’s multiple-comparison test was performed with the SAS version 9.2 software (SAS Institute Inc., Cary, NC, USA). These were used for the statistical analysis in this study. *p* < 0.05 was considered statistically significant. The results were expressed as mean ± standard deviation (SD).

## 3. Results and Discussion

### 3.1. Proofing Curves Determined by Rheofermentometer

Figure 1 shows the proofing curves of the CS–HPMC gluten-free dough, including dough height (Figure 1A) and gas production (Figure 1B) kinetics. The dough height and gas production curves rapidly escalated at the beginning of proofing (0–30 min), and exhibited the greatest height and gas production capacity at 30 min. After this, dough experienced obvious decline with proofing time extended to 70 min. The dough height and gas production capacity maintained constant from 70 min to the end of proofing.

In this study, based on the dough height kinetics, three phases could be classified for the 180 min proofing process, i.e., rapidly rising phase at 0–30 min, slowly decreasing phase at 30–70 min, and the steady state at 70–180 min. The dough matrix provided a substrate for the yeast metabolism in the form of soluble carbohydrates (mainly the fermentable mono- and disaccharides) at the early stage of proofing, causing a quick inflation of dough by CO_2_ [17]. When the soluble carbohydrate in the dough was depleted along with proofing, the metabolism of the yeast slowed down, resulting in an obvious decline in gas production capacity. Different from gluten-containing dough, the phase of dough collapse phenomena was not observed in CS–HPMC gluten-free dough [22]. This might be attributed to the fact that HPMC replaced gluten in gluten-free dough, which was not sensitive to the acids and reducing agents (e.g., butyrate and glutathione, etc.) produced by yeast [20]. In order to clarify the variation of gluten-free dough during proofing, three points (CS–HPMC gluten-free doughs proofing for 30, 60, and 90 min) distributed in three phases were selected to monitor the variation of soluble carbohydrate composition, moisture distribution, and rheology as follows.

### 3.2. Changes of the Soluble Carbohydrate Composition

The effect of proofing on the soluble carbohydrate composition in CS–HPMC gluten-free dough is shown in Figure 2. Four monosaccharides were detected as the main composition of soluble carbohydrates in the CS–HPMC gluten-free dough, including arabinose, glucose, fructose, and mannose. In the initial D-0 dough, glucose content was the greatest (45.60 g/kg), followed by arabinose (16.40 g/kg), and fructose (10.83 g/kg), and mannose (1.40 g/kg) showed the lowest content (1.40 g/kg). Along with proofing, all monosaccharide levels decreased significantly. After proofing for 30 min, glucose content decreased by 40.72%, exhibiting the greatest reduction among all the monosaccharides. Arabinose level decreased by 77.84% when proofing from 30 min to 60 min, which reduced more obviously than the first 30 min (15.24%).

Yeast tends to rapidly consume soluble carbohydrates instead of hydrolyzing starch to provide energy for growth in the initial stage [23]. In this study, the starch structure (crystallinity value) had no significant change (as shown in Appendix A), whereas the four monosaccharide contents decreased with the extension of proofing time. This indicated that in the maximum 90 min proofing period, metabolism and growth of the yeast mainly utilized the soluble carbohydrates of the CS–HPMC gluten-free dough. In particularly, the faster consumption of glucose than the other kind of monosaccharides (especially arabinose) suggested that yeast growth consumed glucose first. The consumed soluble carbohydrates were bio-transformed into cytosolic polysaccharides, peptidoglycans, lipopolysaccharides, and exopolysaccharides, etc. [24]. This phenomenon could contribute to the variation of moisture mobility or distribution, as well as the rheology of the CS–HPMC gluten-free dough [23,25].

### 3.3. Water Morphology

Moisture played a vital role in the formation of dough structure, and its variation and mobility during proofing led to the rheological changes of the dough [26]. Three types of moisture were categorized in the food matrix: non-freezable water was recognized as the part of water strongly bound to the hydrophilic groups, which was characterized by immobility and unfrozen under subfreezing temperatures [27]; freezable water was loosely bound to molecules in the food matrix, characterized by frozen and melted at a temperature lower than 0 °C [28]; free water was mobile and unbound to the food matrix, characterized by crystallized to ice crystal with melting and freezing points at 0 °C [29]. Moisture variations in CS–HPMC gluten-free dough could be investigated through the determination of freezable and non-freezable water contents using a differential scanning calorimetry (DSC).

The effect of proofing on the freezable and non-freezable water contents in CS–HPMC gluten-free doughs were summarized in Table 1. With the extension of proofing, no apparent variations in melt enthalpy (ΔH) and freezable water content (C_F_) were found, while non-freezable water content (C_NF_) significantly decreased from 44.24% to 41.39%. The obvious decrease of C_NF_ suggested that proofing reduced the strongly bounded water proportion and raised water activity in the CS–HPMC gluten-free doughs. On the one hand, the proliferation of the yeast during proofing converted some bound water into available free water, which could increase water activity [30]. On the other hand, the hydrolysis of the soluble carbohydrates by the yeast contribute to the release of a portion of bound water, as discussed in Section 3.1. The slight increase of ΔH might be attributed to the reduction of the total moisture content (C_F_ + C_NF_), which interfered with the starch gelatinization and improved their heat absorption [31].

### 3.4. Water Mobility

To explore the mechanism of how the proofing influences the rheology and structure of gluten-free doughs, water mobility in CS–HPMC dough was performed by low-resolution ^1^H NMR to record the variation of the T_2_ relaxation times.

The effect of proofing on the spin–spin relaxation time of CS–HPMC gluten-free doughs is shown in Figure 3. Three spin–spin relaxation time constants have been identified, namely, T_21_ as the first relaxation time, T_22_ as the second relaxation time, and T_23_ as the third relaxation time. Table 2 shows the spin–spin relaxation time (T_21_, T_22_, and T_23_) and the amplitudes of the three components (A_21_, A_22_, and A_23_). The existence of these time constants indicated that there were two or three fractions of water with different relaxation rates in the CS–HPMC gluten-free doughs [32]. T_21_ and T_22_ of the original D-0 dough were 4.50 and 31.44 ms, which, respectively, increased to 4.82 and 41.50 ms after proofing for 90 min. With the proofing time prolonged from 0 to 60 min, the third relaxation time (T_23_) appeared at 2171.12 ms, which moved forward to 766.4 ms after proofing for 90 min. A_21_ of the native D-0 were 18.52%, which increased to 24.00% with the raise of the proofing time, whereas A_22_ decreased from 81.48% to 75.84%. As for D-60 and D-90, A_23_ increased from 0.03% to 0.19%.

T_2_ distributions (two distinct regions), including T_21_ and T_22_, were usually observed in doughs, ranging from 2 to 5 ms and from 10 to 100 ms, respectively [26]. In the present study, the two T_2_ distributions, T_21_ (2–5 ms) and T_22_ (10–100 ms), were detected as well in the CS–HPMC gluten-free doughs. Because the CS and HPMC content was constant, low-resolution ^1^H NMR results could reflect the effect of proofing on the water distribution of the gluten-free doughs. T_21_ distribution corresponds to the part of water tightly bound to starch, whereas T_22_ distribution corresponds to the part of water on the hydration sites of HPMC and the surface of starch [33,34]. The increase of T_21_ and A_21_ suggested a decrease in the bound tightness between water and dough. To some extent, this might reflect the hydrolysis of soluble carbohydrates by microbial metabolic enzymes during proofing. As for the decreased A_22_ with the increased T_22_, proofing improved water mobility, but also consumed part of the free water involved in the hydration sites of HPMC and CS for proliferation. For D-60, the new T_23_ distribution was the same as that of free water, which has a T_2_ of 2–3 s [26]. This might result from the rapid consumption of soluble carbohydrates and proliferation of the yeast in the dough, which contribute to the release of the bound water and the improvement of the water activity [25]. As the proofing time increased, T_23_ reduced to smaller than that of free water. The increase of A_23_ with the decreased of T_23_ might be attributed to the consumption of free water by the yeast, as well as the absorption of the microbial metabolisms, such as exopolysaccharides, which reduced water mobility [31]. 

### 3.5. Dynamic Rheological Characterization

#### 3.5.1. Dynamic Strain Sweep

The effect of proofing on the rheological properties of the gluten-free dough was evaluated by the viscoelastic changes. As established by the strain sweep experiments, the linear viscoelastic ranges (LVRs) for the CS–HPMC doughs are shown in Table 3. The original D-0 dough presented the lowest LVR limit (0.066%), along with the proofing time increased to 90 min, the LVRs significantly increased to 0.123%.

Corn doughs composed of maize flour, starch, gluten, and zein exhibited LVRs at a strain level of 0.1–0.25% [35,36,37]. In comparison, the CS–HPMC gluten-free doughs exhibited lower LVR limits, which could be attributed to the lack of network structures formed by gluten or zein, which was continuous and stable compared with the CS–HPMC gluten-free dough [16]. In the CS–HPMC dough matrix, the hydrated HPMC formed colloids and attached on the surface of CS granular, and the increased adhesion among granules enhanced the viscosity characteristics of the gluten-free dough. However, the viscosity remained relatively constant at low strain and significantly decreased when the strain increased, which is a shearing-thinning phenomenon that exists in the majority of doughs [38]. In general, starch structure modification (such as gelatinization), and the exopolysaccharide production (play a role as lubricant) during proofing/fermentation were the main reasons for shear-thinning behavior [36]. In this study, the unchanged structure (the crystallinity value without change, as shown in Appendix A) of the CS during proofing exerted no effect on the shear-thinning effect. Instead, the improved LVRs was mainly attributed to the consumption of soluble carbohydrates in the CS–HPMC dough, reducing the lubrication action and lessened shear-thinning behavior [31].

#### 3.5.2. Dynamic Frequency Sweep

Dynamic rheological measurements permit the determination of materials without structure damaging. To explore the effect of proofing on the dough structure in CS–HPMC gluten-free dough, a dynamic frequency sweep was conducted, and the results are shown in Table 3. Frequency (ω) varies at 0.1–100%, while the strain remains constant at 0.5%.

Based on the frequency sweep experiments, all the CS–HPMC gluten-free doughs had a greater storage modulus (G′) than the loss modulus (G″) over the whole frequency range, indicating the elastic behavior. The logarithmic plot of log G′ = z′ log ω + K [39] can be used to evaluate the dependence of G′ on frequency, which could reflect the changes of molecular interactions in CS–HPMC doughs. In this logarithmic plot, z′ value more close to 0, the more obvious the frequency independence, suggesting a more stable dough microstructure with strong intermolecular interactions (such as covalent linkage). Whereas, the greater the z′ value, the more dependent the frequency, indicating a less stable microstructure under weak molecular interactions (such as molecular entanglement) of the dough. The K values record the strength of the doughs, with a higher K value indicating higher strength of the dough [40]. The goodness-of-fit was evaluated by means of the corresponding coefficients of determination (*R*^2^).

Table 3 shows z′ > 0, indicating that all the CS–HPMC doughs were frequency-dependent with weak molecular force, which was in agreement with most gluten-free dough based on different raw materials and additives [41,42]. With the extension of proofing time, z′ values increased from 0.157 to 0.190, signifying the increased frequency-dependence and decreased molecular interactions in the dough [43]. This might be ascribed to the fact that the consumption of soluble carbohydrates decreased the hydrogen bonding and molecular entanglements with starch and HPMC in dough. In addition, the effect of yeast on dough rheology depends on the production of specific metabolites produced during proofing, especially ethanol and succinic acid, which decreased dough stability and extensibility [44]. Furthermore, the decrease of C_NF_ and increase of T_21_ and T_22_, as well as the appearance of T_23_ (after 60 min proofing), indicated that dough compositions tended to bio-transform from bound state to free ones during proofing, reducing the stability of dough microstructures.

#### 3.5.3. Temperature Sweep

The temperature sweep monitors changes of the dough microstructure during the cooking course, especially for the gelatinization of CS in the dough. |G^*^| is the absolute value of the vector between G′ and G″, and Tan δ is the ratio of G″ to G′. Dough with higher |G^*^| and tan δ indicates higher intensity and viscosity, respectively. In this study, Tan δ < 1 indicated a predominance of elastic over viscous properties in the CS–HPMC dough.

The effect of proofing on the complex shear modulus |G^*^| and loss factor tan δ of CS–HPMC gluten-free dough during the dynamic temperature sweep is shown in Figure 4. The increase of |G^*^| during heating are essentially due to starch gelatinization in the CS–HPMC dough. Along with the temperature increase from 25 to 60 °C, |G^*^| slightly decreased for the water swelling of the CS and HPMC, leading to an increase of dough volume and an initial softening of the dough [45]. |G^*^| rapidly increased along with the temperature rising from 60 to 90 °C, indicating the beginning of starch gelatinization. |G^*^| reached the peak (|G^*^|_max_) and then decreased significantly in the process of the temperature being raised from 60 to 90 °C. In this phase, |G^*^| decreased along with the extension of proofing time. This might be attributed to the decomposition of soluble carbohydrates, as well as the improved water mobility in the dough, which decreased dough density.

Tan δ rose along with the temperature increasing, and after reaching the peak (tan δ_max_), it sharply declined. Tan δ_max_ is usually used as a tool to identify the onset of starch gelatinization, whereas |G^*^|_max_ can be used as a tool to identify the peak gelatinization [46]. |G^*^|_max_ and tan δ_max_ at different temperatures, denoted by T_|G*|max_ and T_tan δmax_, increased significantly along with the extension of proofing time (*p* < 0.05). Although the soluble carbohydrates were decomposed, and the water mobility improved during proofing, yeast proliferation still likely restrained a great part of the free state water to achieve appropriate A_w_, which prohibits water access to amorphous parts of the starch granules and causes an increase of the start and peak gelatinization temperatures [31].

### 3.6. Creep and Recovery Measurements

Creep and recovery measurements were also made on the CS–HPMC gluten-free doughs. The creep-recovery curves of CS–HPMC gluten-free doughs (as shown in Appendix A) exhibited typical viscoelastic characteristics, which was consistent with the majority of doughs [47].

Table 4 shows the effect of proofing on the creep-recovery parameters. The maximum creep strain (J_max_) reflected dough rigidity: the higher the J_max_ value, the smaller the dough deformation [48]. The zero shear viscosity (η_0_) presented the dough’s flowability, which may provide some insight into dough macrostructure. The higher relative elastic part of maximum creep compliance (J_e_/J_max_) indicated the higher recovery capacity of the dough [48], whereas the higher relative viscosity part of maximum creep compliance (J_v_/J_max_), the better stability of the dough to trap gas [49]. Along with the extension of proofing time, J_max_ and J_v_/J_max_ significantly increased from 2.64 × 10^−5^ 1/Pa and 78.60% to 7.56 × 10^−5^ 1/Pa and 84.54%, respectively, whereas η_0_ and J_e_/J_max_ obviously reduced from 1.82 × 10^6^ Pa·s and 21.40% to 0.74 × 10^6^ Pa·s and 15.46%, respectively. This suggested that proofing raised dough rigidity and gas trapping capacity, but reduced the flowability and recovery capacity of the dough.

In the CS–HPMC gluten-free dough, starch granules adhered to one another in the presence of the hydrated HPMC, and their mobility depends on the hydration of the HPMC [41,46]. In the process of dough proofing, yeast growth competed water with starch and HPMC, resulting in declined flowability and increased rigidity. In addition, the formation of gas bubbles in the dough interrupted the elastic dough structure and led to a highly viscous dough response [30].

## 4. Conclusions

The effect of proofing on the rheology and moisture distribution of corn starch-hydroxypropylmethylcellulose (HPMC) gluten-free dough was investigated. Along with proofing, the bound water proportion reduced, whereas the water mobility improved. In addition, the shear-thinning behavior, molecular interactions, flowability, and recovery capacity of the gluten-free dough decreased, while the dough rigidity and gas trapping capacity improved. These changes could be attributed to the decreased interactions, including hydrogen bonding and molecular entanglements, in the gluten-free dough. Furthermore, yeast growth competed water with starch and HPMC, resulting in declined flowability and increased rigidity. This study could supplement an understanding on the rheological variation, as well as the moisture distribution of the gluten-free dough during proofing, contributing to the development of gluten-free breads with improved quality and consistency.

## Figures and Tables

**Figure 1 foods-12-00695-f001:**
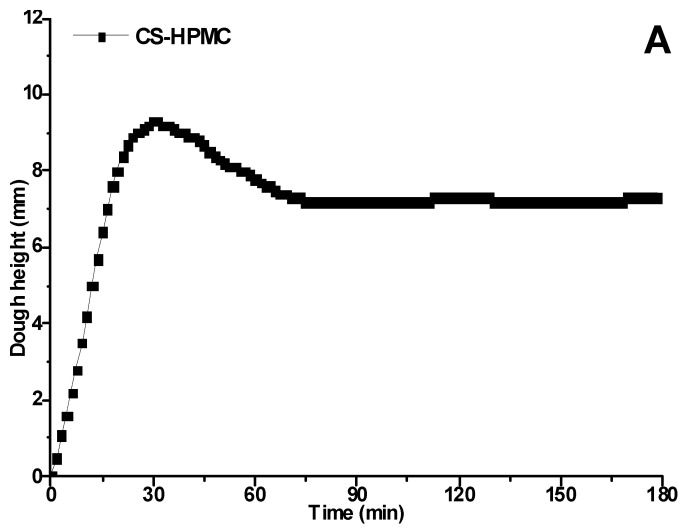
Proofing curves of the CS–HPMC gluten-free dough. Dough height (**A**) and gas production kinetics (**B**).

**Figure 2 foods-12-00695-f002:**
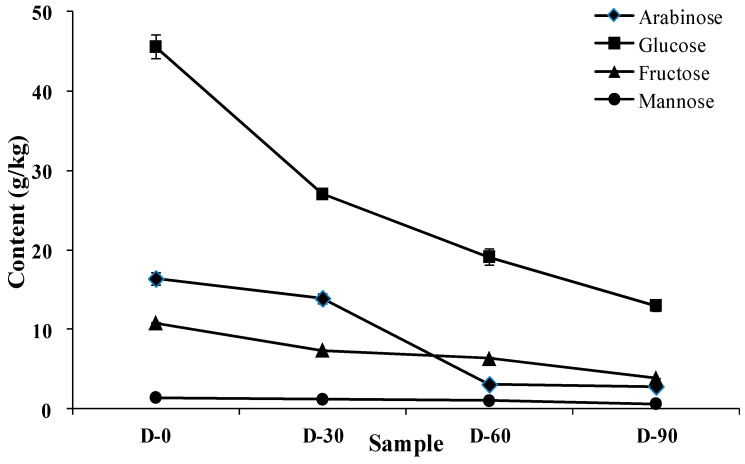
Effect of proofing on the soluble carbohydrate composition (g/kg) of CS–HPMC gluten-free dough.

**Figure 3 foods-12-00695-f003:**
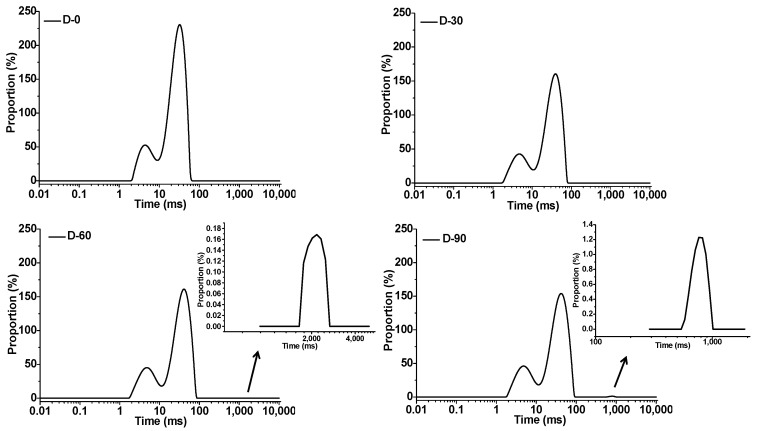
Effect of proofing on the continuous distribution of spin–spin relaxation time of CS–HPMC gluten-free dough.

**Figure 4 foods-12-00695-f004:**
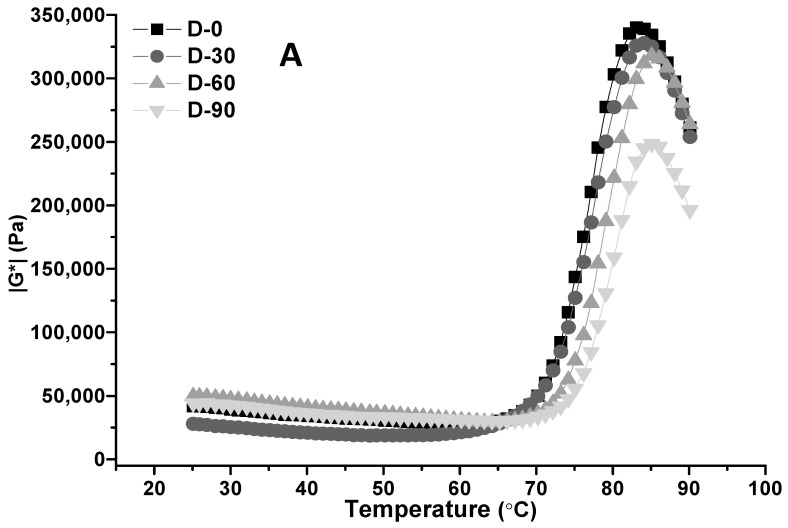
Effect of proofing on the complex shear modulus |G^*^| (**A**) and loss factor tan δ (**B**) of CS–HPMC gluten-free dough during the dynamic temperature sweep.

**Table 1 foods-12-00695-t001:** Effect of proofing on the melt enthalpy, freezable water and non-freezable water contents in CS–HPMC gluten-free dough.

Sample	ΔH (J/g)	C_F_ (%)	C_NF_ (%)
D-0	91.72 ± 1.75 a	27.46 ± 0.53 a	44.24 ± 0.53 a
D-30	92.12 ± 2.01 a	27.58 ± 0.71 a	43.92 ± 0.71 a
D-60	92.80 ± 0.87 a	27.78 ± 0.26 a	42.72 ± 0.26 ab
D-90	93.96 ± 0.25 a	28.13 ± 0.15 a	41.39 ± 0.15 b

Data followed by the same letter in the same column are not significantly different (*p* > 0.05). ΔH (J/g): melt enthalpy; C_F_ (%): freezable water content; C_NF_ (%): non-freezable water content.

**Table 2 foods-12-00695-t002:** Effect of proofing on the relaxation times, T_2_, and populations of CS–HPMC gluten-free dough.

Sample	T_21_	T_22_	T_23_	A_21_	A_22_	A_23_
D-0	4.50 ± 0.00 b	31.44 ± 0.00 c	-	18.52 ± 0.06 d	81.48 ± 0.14 a	-
D-30	4.82 ± 0.00 a	38.72 ± 0.21 b	-	21.86 ± 0.17 c	78.14 ± 0.32 b	-
D-60	4.82 ± 0.00 a	41.50 ± 0.00 a	2171.12 ± 0.30 a	23.16 ± 0.20 b	76.82 ± 0.09 c	0.03 ± 0.01 b
D-90	4.82 ± 0.00 a	41.50 ± 0.00 a	766.34 ± 0.00 b	24.00 ± 0.32 a	75.84 ± 0.27 d	0.19 ± 0.01 a

Data followed by the same letter in the same column are not significantly different (*p* > 0.05). “-” icon indicated no data detected. T_21_ (ms): the first relaxation time of T_2_; T_22_ (ms): the second relaxation time of T_2_; T_23_ (ms): the third relaxation time of T_2_; A_21_ (%): the amplitudes of T_21_; A_22_ (%): the amplitudes T_22_; A_23_ (%): the amplitudes of T_23_.

**Table 3 foods-12-00695-t003:** Effect of proofing on the structure stability of CS–HPMC gluten-free dough.

Sample	LVR (%)	z′	K	*R* ^2^
D-0	0.066 ± 0.003 c	0.157 ± 0.008 b	4.861 ± 0.039 a	0.912 ± 0.004 b
D-30	0.075 ± 0.004 c	0.166 ± 0.002 ab	4.805 ± 0.026 ab	0.999 ± 0.000 a
D-60	0.104 ± 0.003 b	0.188 ± 0.012 a	4.810 ± 0.018 ab	0.992 ± 0.006 a
D-90	0.123 ± 0.005 a	0.190 ± 0.004 a	4.732 ± 0.032 b	0.999 ± 0.000 a

Data followed by the same letter in the same column are not significantly different (*p* > 0.05). LVR: linear viscoelasticity region; z′: the degree of dependence of G′ on frequency sweep; K′: the strength of molecular interactions; *R*^2^: the corresponding coefficients of determination.

**Table 4 foods-12-00695-t004:** Effect of proofing on the deformation resistance and recovery capacities of CS–HPMC gluten-free dough.

Sample	Creep Phase		Steady State	
	J_max_ × 10^5^ (1/Pa)	η_0_ × 10^−6^ (Pa·s)	J_e_/J_max_ (%)	J_v_/J_max_ (%)
D-0	2.64 ± 2.10 d	1.82 ± 0.52 a	21.40 ± 1.37 a	78.60 ± 2.34 c
D-30	3.20 ± 1.05 c	1.69 ± 0.25 a	19.44 ± 0.91 ab	80.56 ± 1.95 ab
D-60	5.46 ± 1.28 b	0.79 ± 0.04 b	18.77 ± 1.22 b	81.23 ± 2.07 ab
D-90	7.56 ± 1.21 a	0.74 ± 0.11 b	15.46 ± 1.79 c	84.54 ± 1.53 a

Data followed by the same letter in the same column are not significantly different (*p* > 0.05). J_max_ (1/Pa): the maximum creep compliance; η_0_ (Pa·s): the zero shear viscosity, J_e_/J_max_ (%): the relative elastic part of maximum creep compliance; J_v_/J_max_ (%): the relative viscous part of maximum creep compliance.

## Data Availability

The data are available from the corresponding author.

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
