# Peer review of "Effect of Proofing on the Rheology and Moisture Distribution of Corn Starch-Hydroxypropylmethylcellulose Gluten-Free Dough"

_foods, 2023, doi:10.3390/foods12040695_

Round 1

Reviewer 1 Report

The manuscript with the title: "Rheology and Moisture Distribution of Corn starch-Hydroxypropylmethylcellulose Gluten-free Dough: Effect of Proofing" is about the effect of proofing on the rheology and moisture distribution of corn starch-hydroxypropylmethylcellulose (HPMC) gluten-free dough. In general, the research is interesting, but the presentation of the research is not professional. I recommend the author rewrite the manuscript and improve it substantially. The comments on the manuscript are as follows:

1- Title: a good title should clearly state the "Why" and "How" of research. You need to rewrite it to show "Why" you are going to do the research.

2- Abstract: You started the abstract with "We", but this article has only 1 author. Recommend rewriting all sentences as "Passive", not "Active". Also, elements of the abstract are missing. You need to have, objective(s), Methodology, Main results, and Conclusion in your abstract.

3- Introduction: Weak introduction, you need to clearly mention about problem statement, the significance of the study and your originality.

4- Methodology: Weak and needs to improve. For example in line 78, again you mentioned "According to our previous study" this is not the right way for writing methodology.

5- Results and discussion: Improve the discussion part, it is not a deep discussion, frequently you have used: "According to previous reports", which is not a good feeling for authors.

6- Conclusion: Make it short and informative. In conclusion, you need to justify the hypothesis and not repeat your results.

Author Response

Dear editor, 

Submission of the revised manuscript entitled “Effect of Proofing on the Rheology and Moisture Distribution of Corn starch-Hydroxypropylmethylcellulose Gluten-free Dough”.

Your comments on the above manuscript are well appreciated. We have revised the manuscript based on your suggestions. You can find the corrections made in the list attached to this letter.

Thank you for your favorable consideration.

Yours sincerely,

Duqin Zhang

Institute of Cereal & Oil Science and Technology, Academy of National Food and Strategic Reserves Administration, Beijing, China

Editor:

(I) Please check that all references are relevant to the contents of the manuscript.

Response: The references were carefully checked according to the editor’s suggestion, and confirmed to be relevant to the contents of the manuscript.

(II) Any revisions to the manuscript should be marked up using the “Track Changes” function if you are using MS Word/LaTeX, such that any changes can be easily viewed by the editors and reviewers.

Response: “Track Changes” function was used to mark up any changes in the revised manuscript according to the editor’s suggestion. Please see the revised manuscript for details.

(III) Please provide a cover letter to explain, point by point, the details of the revisions to the manuscript and your responses to the referees’ comments.

Response: This cover letter was provided to explain the details of the revisions to the manuscript according to the editor’s suggestion.

#Reviewer 1

The manuscript with the title: "Rheology and Moisture Distribution of Corn starch-Hydroxypropylmethylcellulose Gluten-free Dough: Effect of Proofing" is about the effect of proofing on the rheology and moisture distribution of corn starch-hydroxypropylmethylcellulose (HPMC) gluten-free dough. In general, the research is interesting, but the presentation of the research is not professional. I recommend the author rewrite the manuscript and improve it substantially. The comments on the manuscript are as follows:

1- Title: a good title should clearly state the "Why" and "How" of research. You need to rewrite it to show "Why" you are going to do the research.

Response: The tile has been changed to “Effect of Proofing on the Rheology and Moisture Distribution of Corn starch-Hydroxypropylmethylcellulose Gluten-free Dough” according to the reviewer’s suggestion. Please see the revised manuscript.

2- Abstract: You started the abstract with "We", but this article has only 1 author. Recommend rewriting all sentences as "Passive", not "Active". Also, elements of the abstract are missing. You need to have, objective(s), Methodology, Main results, and Conclusion in your abstract.

Response: The abstract has been modified according to the reviewer’s suggestion. Please see the revised manuscript for details.

3- Introduction: Weak introduction, you need to clearly mention about problem statement, the significance of the study and your originality.

Response: The significance of the study and the originality has been supplemented in the introduction according to the reviewer’s suggestion. Please see line 58-63 in the revised manuscript for details.

4- Methodology: Weak and needs to improve. For example in line 78, again you mentioned "According to our previous study" this is not the right way for writing methodology.

Response: The sentence has been modified according to the reviewer’s suggestion, please see line 78-80, and 144-146 in the revised manuscript for details.

5- Results and discussion: Improve the discussion part; it is not a deep discussion, frequently you have used: "According to previous reports", which is not a good feeling for authors.

Response: The discussion part has been improved according to the reviewer’s suggestion. Please see line 239-240, 253-256, 303-304, 335-336, 344-346, and 375-378 in the revised manuscript for details.

6- Conclusion: Make it short and informative. In conclusion, you need to justify the hypothesis and not repeat your results.

Response: Conclusion has been rewritten to make it more concise and informative according to the reviewer’s suggestion. Please see line 445-450 in the revised manuscript for details.

#Reviewer 2

1- The equations set forth in lines 129 and 160 should be numbered.

Response: We have numbered all the equations according to the reviewer’s suggestion. Please see line 129, 160, 181, 186, 190, and 191 in the revised manuscript for details.

2- In line 128, it is written ¨dato¨ instead of data.

Response: We have modified “data” to “dato” according to the reviewer’s suggestion. Please see line 159 in the revised manuscript for details.

3- The references used in lines 95, 104, 122, 213 and 360 have two formats. It is suggested to use only one of them.

Response: We have modified the format according to the reviser’s suggestion. Please see line 95, 104, 122, 213 and 360 in the revised manuscript for details.

4- Line 218 reads ¨(…) obvious decline gas production capacity¨; the following grammatical modification is suggested: ¨(…) obvious decline in gas production capacity¨.

Response: We have corrected “(…) obvious decline gas production capacity” into “(…) obvious decline in gas production capacity” according to the reviewer’s suggestion. Please see line 218 in the revised manuscript for details.

5- Line 237 reads ¨(…) reduce more obvious than frist…¨; the following change is suggested: ¨(…) reduce more obviously than frist…¨ for correct grammar.-

Response: We have modified “obvious” to “obviously” according to the reviewer’s suggestion. Please see line 238 in the revised manuscript for details.

6.-In section 2.4 we talk about the HPLC technique, for the identification and quantification of the four monosaccharides, for which it is suggested to show the chromatograms or at least include it in the supplementary material; This, in order to see the resolution between the peaks of glucose, fructose and mannose, already have close retention times in value.

Response: We have supplemented the HPLC chromatogram of the CS–HPMC gluten-free dough according to the reviewer’s suggestion. Please see the “Supplementary Figure 1” in the revised Supplementary data for the details.

7- It is suggested to show the flow-recovery curves in section 3.6 or their inclusion in the supplementary material.

Response: We have supplemented the creep-recovery curves of CS–HPMC gluten-free doughs according to the reviewer’s suggestion. Please see the “Supplementary Figure 2” in the revised Supplementary data for details.

8- In the declaration of the complementary material, 3 elements are declared: a figure, a table and a video. However, only the table is included, the only element referenced in the body of the article (line 241-242).

Response: We have supplemented two figures (Supplementary Figure 1 and 2) in the complementary material, and referenced in the body of the article. Please see the revised Supplementary data and line 117-118 and 416-417 in the revised manuscript for details.

Reviewer 2 Report

1- The equations set forth in lines 129 and 160 should be numbered.
2- In line 128, it is written ¨dato¨ instead of data.
3- The references used in lines 95, 104, 122, 213 and 360 have two formats. It is suggested to use only one of them.
4- Line 218 reads ¨(…) obvious decline gas production capacity¨; the following grammatical modification is suggested: ¨(…) obvious decline in gas production capacity¨.
5- Line 237 reads ¨(…) reduce more obvious than frist…¨; the following change is suggested: ¨(…) reduce more obviously than frist…¨ for correct grammar.- 6.-In section 2.4 we talk about the HPLC technique, for the identification and quantification of the four monosaccharides, for which it is suggested to show the chromatograms or at least include it in the supplementary material; This, in order to see the resolution between the peaks of glucose, fructose and mannose, already have close retention times in value.
7- It is suggested to show the flow-recovery curves in section 3.6 or their inclusion in the supplementary material.
8- In the declaration of the complementary material, 3 elements are declared: a figure, a table and a video. However, only the table is included, the only element referenced in the body of the article (line 241-242).

Author Response

(The authors gave the same response as above.)

Round 2

Reviewer 1 Report

Author fairly responded to all comments.